# Perception of Medical Humanities among Polish Medical Students: Qualitative Analysis

**DOI:** 10.3390/ijerph20010270

**Published:** 2022-12-24

**Authors:** Marta Makowska, Agnieszka J. Szczepek, Inetta Nowosad, Anna Weissbrot-Koziarska, Joanna Dec-Pietrowska

**Affiliations:** 1Department of Economic Psychology, Kozminski University, Jagiellońska 57, 03-301 Warszawa, Poland; 2Faculty of Medicine and Health Sciences, University of Zielona Góra, 65-046 Zielona Góra, Poland; 3Department of Otorhinolaryngology, Head and Neck Surgery, Charité-Universitätsmedizin Berlin, Corporate Member of Freie Universität Berlin and Humboldt-Universität zu Berlin, 10117 Berlin, Germany; 4Faculty of Social Sciences, University of Zielona Góra, 65-046 Zielona Góra, Poland; 5Faculty of Social Sciences, University of Opole, 45-040 Opole, Poland

**Keywords:** medical humanities (MH) courses, medical education, medical schools, medical students’ perception university management, focus groups, Poland

## Abstract

Medical humanities (MH) courses are a critical element of the medical curriculum influencing the establishment of a physician in the medical profession. However, the opinion about MH among medical students remains unknown. Interviews from seven focus groups were analysed. The students attended one of three Polish medical schools in Gdansk, Krakow, and Warsaw and were recruited to the discussion focused on the impact of drug manufacturers’ presence at medical universities on socialization in the medical profession. Thematic analysis was conducted using the theoretical framework of social constructivism. The students’ opinions about the MH classes arose during the analysis. In six groups, students thought that MH courses would be helpful in their future medical practice. However, in four groups, different opinion was expressed that MH courses were unnecessary or even “a waste of time”. Factors discouraging students from the MH classes included poorly taught courses (monotonous, uninteresting, unrelated to medical practice, taught by unsuitable lecturers). Secondly, students thought that the time investment in the MH was too extensive. Furthermore, curriculum problems were identified, reflecting the incompatibility between the content of MH courses and teaching semesters. Lastly, some students stated that participation in MH courses should be elective and based on individual interests. Addressing problems recognized in this work could improve the training of future Polish physicians.

## 1. Introduction

The literature defines medical humanities (MH) in various ways [1,2,3], but all definitions acknowledge their multi- and interdisciplinary character. MH integrates psychology, social sciences, arts, history, ethics, and philosophy into one discipline with the goal of better understanding the human side of medicine [4]. The Association for Medical Humanities (UK) clearly defined the MH’s aims as developing communication skills, ethical concerns, critical and reflective thinking, concern for different aspects of the human condition, and own professional and personal values resulting in “intersubjective” knowledge of the patient and physician [4].

The importance of MH in medical education has been known since the 1960s [5,6,7,8,9], and today, MH classes are widespread, providing theoretical and practical frameworks for integrating MH into medicine [10]. “The goals of a medical humanities curriculum are to (1) ingrain aspects of professionalism, empathy, and altruism; (2) enhance clinical communication and observation skills; (3) increase interprofessionalism and collaboration; and (4) decrease burnout and compassion fatigue” [11]. Contemporary research points to the variety of approaches used to teach MH [12] and supports the positive outcomes of MH teaching during undergraduate studies [13], despite the still existing dichotomy between courses teaching practical medical skills and humanity-oriented classes [14]. The positive experience with MH has induced modifications of study programs in other countries, such as Germany [15] and Belgium [16]. In Poland, the reputation of MH has increased, inducing changes in education programs [17,18]. However, Polish medical universities implement MH in a wide variety of ways. At the same time, no universal curriculum model for MH classes exists [19].

Social competencies developed by MH strengthen understanding and reaction to the human behavior of students and medical practitioners during the decision-making pathway (e.g., facilitating the diagnosis process). They also strengthen cooperation skills, positively changing the environment and enhancing the therapy outcomes [20]. MH-enhanced education should prevent professional burnout and moral erosion [21]. As a result, the quality and efficiency of physicians’ work improve in many areas of functioning.

In Poland, MH represents a vital education area and is a subject of social discussions in various environments, from scientific to management groups. The Polish Provosts’ Foundation created a report, “Education for the future—the quality of education”, which pointed to the critical question that worries stakeholders: How to educate students during times of unpredictable changes? [22]. A “future” is difficult to define, and the report’s authors express their opinions on a very general level. They suggest the importance of finding sustainable elements in the transferred knowledge and skills. They also recommend preparing students for change by developing adaptation and management expertise. Lastly, they advise education in social competencies as a sustainable element of education, preparing for flexibility in dealing with the ever-changing social environment [22]. Research conducted in Poland indicated the need to offer courses focusing on the humanization of medicine and medical communication during early medical education [23]. Some Polish medical universities have implemented the necessary changes in their curricula and assessed their effectiveness [24].

Ultimately, the beneficiaries of the MH program in medical schools are the students. However, their opinions regarding the need for MH classes, the value of the curriculum, and the quality of program execution are unknown. Given that the success of MH classes depends on students’ accepting the new approach [25], the present work aims to answer the question: *What is medical students’ opinion about MH*?

## 2. Materials and Methods

### 2.1. Setting

Nine focus group interviews were conducted in November 2019 and involved medical students from three large Polish cities—Gdansk, Krakow, and Warsaw. Students from only one medical school per city were recruited. The main objective of the study was to identify the influence of pharmaceutical companies on socialization in the medical profession (see published papers [26,27]); however, a substantial part of the conversation was devoted to socialization in the medical profession in general. To create a friendly and relaxed atmosphere, the moderator asked introductory questions, for example, about the pros and cons of studying medicine.

The sampling for the study was purposive. The recruiting company invited second-year or higher medical students to participate. There were 68 students in the seven analyzed in this article focus groups. In Gdańsk, there were 26 persons recruited; in Krakow, 16 (8 women and 8 men); and in Warsaw, 26 (11 women and 15 men).

### 2.2. Framework

The theoretical framework in this work was social constructivism. It is based on certain assumptions: (1) reality is created by human actions, and it cannot be discovered because it is socially created; (2) knowledge is a human product, and it is socially and culturally created; (3) learning is a social process, and meaningful learning takes place when a person is engaged in social activities [28]. Thus, social constructivism assumes that people construct their knowledge while interacting with others [29]. Things that seem neutral or ordinary in a society, such as a gender and race, are socially constructed and do not reflect reality. How we perceive the world depends on the language, culture, or historical epoch [30]. Therefore, the medical students were characterized by their unique constructs recognized by the research method. Focus groups share some assumptions with social constructivism. The subjects construct their own framework to give meaning to their experiences. That framework can change during discussion (interaction with others), creating new knowledge [31].

### 2.3. Qualitative Analysis

All focus groups lasted approximately 2 h. They were all run by the same moderator, who also acted as a coder and used the same script. Based on the recordings (audio and video), an anonymized transcript of all focus groups was prepared. Thematic analysis was used to examine the result [32,33,34]. The thematic analysis involves searching the data set to identify, analyze and report recurring patterns [35] and can be applied to various paradigms. However, it seems particularly well suited to study social constructivism, on which this research initially focused [36].

As an a priori hypothesis was not developed, codes and themes were developed inductively from the data. The data was coded by the project’s PI, who participated in the entire study process and interpreted the context. The thematic analysis followed the steps described in the literature: familiarization, identifying themes, indexing, charting, and interpretation [37,38]

Ethics committee approval was waived for this study.

## 3. Results

During the thematic analysis, two main themes were distinguished: (1) the MH as an element of medicine; (2) the discouragement of learning MH. In the second theme, the following subthemes were categorized: (a) poor quality of teaching, (b) time-consuming; (c) elective versus compulsory format; (d) allocation to inappropriate study years (Figure 1).

### 3.1. The MH as an Element of Medicine

In all analyzed groups, some students considered medical studies to be complex and requiring much dedication. Comparing themselves to students of other disciplines, medical students often indicated that due to a large amount of study, they have much less free time to develop their passions and interests than those studying something else. Some students even put their studies in opposition to humanistic or technological studies, treating their studies more as learning a trade:


*Our studies are trade learning, and there is not much of an academic element, as is the case when someone takes MH or social studies, and there is also no creative work (…) students of, for example, technical faculties are obliged to do a vast number of projects (…) [medicine] is simply learning a trade.*


However, some students believed that their studies combined various fields and that MH was also an element of them:


*(…) medicine is comprehensive; it connects various fields, both humanities and science (…); everything is mixed, it’s not dull.*


There were also contradictory voices in the discussions regarding the need to attend MH courses during medical studies. Some students thought such courses should be discarded from the already overloaded curriculum, and they unflatteringly called them “a waste of time”.


*I would really discard some courses from the first and second years of study; they are*
*a waste of time.*


However, that opinion was not shared by all medical students. Some recognized the positive aspects of learning the MH, and it was not hard for them to find the application of MH courses in their future medical practice. Below is an example of such a discussion.


*Person A: What I like is that we get to know different ways of communication during these studies (…) that we have such classes, a bit on the border of psychology, a bit on the border of communication, and you can actually use it. This is pretty cool. (…)*



*Person B: It seems to me that in some way, we not only learn about human anatomy, physiology, and diseases, but we learn a little bit by superimposition; we learn how to handle them [patients], what they are and about their personal qualities.*


### 3.2. Discouragement to Learning MH

In five of the seven analyzed focus groups, the MH theme arose when the participants discussed the disadvantages of studying medicine (Gdansk, GP 2; Krakow GP 6; Warsaw GP 7, Warsaw GP 8, Warsaw GP 9). In one group, the theme MH arose when the advantages of studying medicine were discussed (Krakow, GP 5); and in another group, when other issues were debated (Gdansk, GP 3). The students often associated the MH with something negative. Four subthemes of discouragement to learning MH theme were recognized, as described below.

#### 3.2.1. Poor Quality of Teaching

Group discussions indicated that the MH courses are sometimes dull and not attractive. It should also be emphasized that some students glorified the MH courses. It also happened that one course at the same university was praised by some students but not by others, which usually depended on the particular teacher and not the MH course. Below is an example of such a discussion:


*Person A: (….) well, these classes were terrible.*



*Person B: I liked them.*



*Person A: Who conducted yours? I had a young girl who was a physician, but sorry; I have to say it—she was just dumb, really (…)*


In one group, an allegation was put forward against the MH teachers as having no idea of what the actual work of a physician looks like, and their advice is often impractical:


*(…) to these patient-related courses, (…) they are a bit imaginary and focused on an idyllic reality that is not true. (…) The teachers assume that the physician has a lot of time to talk to the patient, so there is a golden and unique rule that a patient should never be interrupted, even if he talks about (…) his life during the war or about his grandchildren. He cannot be interrupted (…). Other rules are also out of touch with reality. (…)*


On the other hand, in the same group, someone else pointed out that physicians should not be teaching MH courses. According to this student, one needs expert knowledge to teach some issues competently, and physicians do not necessarily have it:


*It is not the sociologist who teaches us (…) [sociology], but for example, there is a former geriatrician who conducted it. And he didn’t know much about it (…) He said he was interested in it, but somehow he didn’t really know what he was saying (group laughter). (…)*


It also happened that some students did not like the MH classes because their teacher had “distant views” on issues that were important to the students (e.g., sexual identity or preference issues). As a result, there was a problem with communication:


*But it happened very often that the teacher (…) felt offended by the group because, for example, someone said something that was not correct according to her/his views, e.g., (…) “compared to this sick person, normal people they can do something”, I don’t remember exactly what disease it was about, the teacher began to explain for 20 min that we cannot say that the sick is not normal because (…) and a very long, ideological explanation why you cannot do that. (….) Some of the [female] teachers would literally be offended if they were called [coded male name of the profession] and not [coded female name of the profession—rarely used] because she feels like a feminist and so on.*


Poorly conducted, uninteresting and dull courses discourage medical students from attendance. Sometimes the form (lecture/exercises) is not well suited to the content conveyed. Below, we present a quote from a discussion in one of the groups, during which students talked about how they, “contrary to ethics”, avoid MH classes because, although useful, they are conducted in a “terrible” way:


*Person A: There was a list there, there are 300 of us per year, and 20 people were present and signed the list for the entire group (…) (Laughter)*



*Person B: But these classes are impossible to listen to. For example, I must admit that I was there also once because it was my turn, and it was impossible [to listen to], just like that. (…) Often it is the manner of speaking or the form of a lecture at all (…).*


#### 3.2.2. Time-Consuming Courses

Medical students spend much time learning crucial medical information. According to many study participants, the curriculum is overloaded, and they have to learn too many things by heart. Many believe they learn too few practical, useful things for a physician and spend too much time on other classes, including the MH. Some students felt that the obligation to attend them is a waste of time:


*(…) the problem is that sometimes, these items take up too much of our time and workload, or they are just so poorly conducted, and they are not interesting.*


In another group a student stated:


*(…), e.g., a class where we learned how to make injections took only 10 h compared to (…) collecting medical history, on which we spent 30 h. (…)*


#### 3.2.3. Elective Versus Compulsory Format of Courses

Some students believed that the number of compulsory MH courses should be narrowed down. They suggested offering elective courses, giving students a choice to attend.


*Or maybe as for these unnecessary subjects, we actually had many of these MH courses (…) I don’t know, maybe if some of these courses were elective?*


There were also claims that the number of elective MH courses should be expanded to create the possibility of choosing something interesting:


*I would also raise the issue of elective courses also and give them a disadvantage (…), let’s say [I am interested in] history, political science, or something else, but there are no such courses [to choose from].*


#### 3.2.4. Allocation of MH Classes to Inappropriate Study Years

Students pointed out that some MH subjects are planned during inappropriate years of study, making it difficult for many students to appreciate them. Below is an example of such a statement.


*I think that sometimes the classes at [university name] are poorly matched to the appropriate years; for example, in the second year, we had psychology classes on professional burnout. Well, it’s probably better to have them during the fifth or sixth year than during the second.*


## 4. Discussion

We posed a question: *What is medical students’ opinion about medical humanities*? Our study demonstrated a lack of unanimous opinion about MH among medical students in Poland. The opinions diverged between very positive, expressed by students expecting a further extension of the MH courses, and highly negative, expressed by students demanding MH termination. Moreover, we determined problems with the acceptance of MH classes and their implementation (planning and teaching). This divergence in opinion emphasizes the importance of considering the needs and values of medical students when planning MH curricula.

Our first finding of diverse opinions about MH contrasts with the study of Helen et al., in which 98.5% of participating medical students had positive or highly positive opinions about MH [39]. However, there are substantial methodological differences between the study of Helen et al. and ours, as they performed a semi-structured survey, whereas we used a focus group interview.

The second finding of our work was the identification of the discouragement to learning MH among medical students. This discouragement could be subdivided into four subtopics: poor quality of teaching, time-consuming, format issue (elective versus compulsory), and allocation to inappropriate study years.

Knowledge, competencies, and skills in the humanities and social sciences must consider the person as a human being with his or her problems, capable of self-determination, which is particularly important in the case of professions of public trust, of which the medical profession is one. The contemporary physician cannot only possess technical and practical skills, but it is advisable to develop personal and interpersonal skills to create a relationship with the patient and the patient’s family [40]. The quality of teaching such skills depends on the instructor’s knowledge and competencies and should be monitored by each university using evaluation tools (e.g., post-classes quality assessment) [3,41]. In Poland, training the MH instructors is left to the discretion of individual medical schools. However, in light of our findings, universal introductory MH courses for the academic staff could help shape the teachers. Such training involves considerable costs, which might pose a problem, especially since the state-founded universities in Poland often have insufficient financial resources for educating their staff. Still, such investment should be considered because the students want higher quality teaching of MH.

The student’s opinion about MH courses being time-consuming very likely reflects the feeling that such classes are unnecessary in medical education as they take the time needed to study other “important” subjects [3].

The format issue of the MH courses (elective versus compulsory) was addressed in a recent study, which identified medical students’ preference for elective classes [25], agreeing with our present findings. In addition, previously, we identified vast differences between Polish universities in the form of MH classes and the time investment that varied from as much as 216 h of training to as little as 15 h during the entire course of study [19]. That, again, calls for the unification of the MH curriculum between the universities, at least regarding compulsory courses.

The last identified issue was allocating HM classes to inappropriate study years. In agreement with our results, survey-based qualitative research by Petrou et al. that focused on medical students’ opinions about MH found the need for better integration of MH into clinical subjects [20]. Thus, we recommend an adjustment of the existing curricula to accommodate students’ needs and improve the educational and applied outcomes of MH teaching.

Poland is struggling with a massive shortage of physicians, being the lowest in the EU *per capita* [42]. That shortage has recently increased the number of private and state-funded medical schools, many of which have included MH in teaching programs [19]. However, each university designs its curriculum (including MH courses), and there are considerable discrepancies between the medical schools in MH education. In an era of rapid global and demographic changes, technological advances, and the resulting transformation of medical care, there is a high need to train MH competencies at medical universities. Students’ opinions on when, using which format, and by whom they should be trained in MH should be taken seriously by the universities and help the continuous improvement process in medical education.

### 4.1. Study Limitations

A limitation of this study is purposive sampling, which limits generalizability. Another limitation was the availability of the respondents, contributing to the overrepresentation of third-year students. Respondents who had recently begun their medical education did not have a comprehensive view of MH courses as those more advanced in their study, so the design of future research should probably focus on the more advanced students.

### 4.2. Study Strengths

The study’s strength is the focus group technique, used for the first time in Poland to examine the topic. This technique allowed lively discussions among students and the disclosure of important issues related to medical studies.

### 4.3. Future Research

Our study created a starting point for conducting quantitative research on this issue at Polish medical universities. First of all, it shows what issues should be raised, thus, facilitating the creation of a survey questionnaire. Furthermore, our study indicated that students from more advanced years should participate in such a survey, as they have more experience with MH at the university. Two issues highlighted by students (e.g., allocation of MH to inappropriate study years or inappropriate content of format of the classes) suggest a topic for another study, in which analysis curriculum of Polish medical schools could be conducted to determine which type of MH classes and when are taught in Poland to whether the students’ criticism was justified. Students also raised the issue of being taught MH by physicians who often are unfamiliar with the subject. On the other hand, they had reservations that people teaching MH are detached from medical practice. That suggests a focus of another future study on the professional background of MH teachers in Poland.

## 5. Conclusions

The students’ views on whether medicine and medical studies are related to MH and the need to study MH were different among the respondents. Our study, therefore, indicates the need to demonstrate the importance of MH, e.g., presenting a practical application in future medical practice. The present study determined factors discouraging the respondents from learning humanities—these factors included poor quality of teaching, time investment, allocation to inappropriate years of studies, and unsuitable content or format of the classes. The results of this study may create the basis for further quantitative research on how problematic these issues are in Polish medical universities. Moreover, we advocate a need to create a national MH organization to collaborate with others (e.g., the Association for Medical Humanities in the UK) and to advise Polish medical schools on creating and implementing their MH curricula. This organization should include representatives of advanced medical students to acknowledge their opinions in the future. Addressing these issues can help improve the training of future physicians in Poland.

## Figures and Tables

**Figure 1 ijerph-20-00270-f001:**
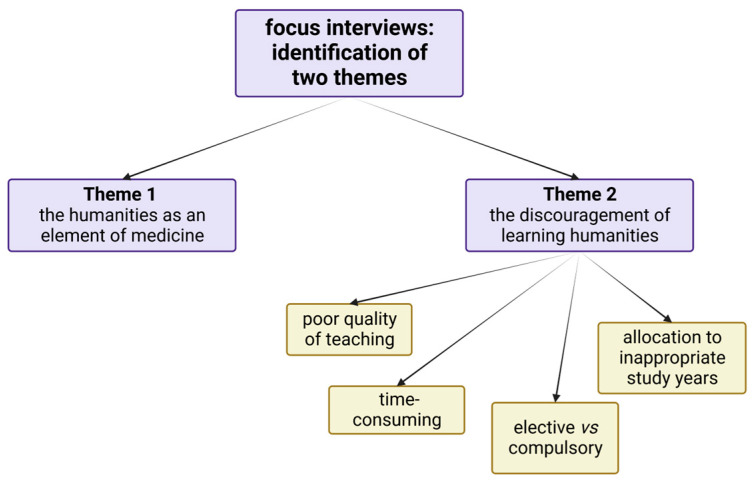
Focus interviews allowed the identification of spontaneous themes and sub-themes. Created with BioRender.com.

## Data Availability

The data from this study were deposited at Social Data Repository https://rds.icm.edu.pl/dataset.xhtml?persistentId=doi:10.18150/UFEA7T (accessed on 23 December 2022).

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
