# Peer review of "Perception of Medical Humanities among Polish Medical Students: Qualitative Analysis"

_ijerph, 2022, doi:10.3390/ijerph20010270_

Round 1

Reviewer 1 Report

Using qualitative analysis of focus groups, the authors explore medical student perceptions of the role of humanities in medical education/practice. The results of this study hold importance for the guiding considerations when developing/implementing humanities education programs for medical students. Some of this importance, however, is lost in the paper as due to confusing language, a scattered structure, poorly described methods, and having much of the introduction and background tangential to the take-home point of this work. Furthermore, the authors repeatedly emphasize the unprompted nature of the humanities theme as a novel factor of this study. While important to mention that it was not explicitly in the interview guide, it should not be part of a repeating theme in the paper. There are several major edits that are required as well as more minor

Title:

1.       When you say “focused interviews,” are you alluding to the focus group or somethings else? it seems like an un-useful descriptor, regardless, and I would suggest removing.

Abstract:

2.       Clarify that the discussion was benefits of medicine (or pharmaceuticals? –mentioned later is paper) and say the theme of MH and MH classes arose during analysis.

3.       Ln 18: Awkward sentence, phrase sounds like you analysed the students rephrase to “Interviews from seven focus groups were analysed…” or “The perceptions of medical students form seven focus groups were analysed…”

4.       Ln 20-21: Simplify sentence to state “Thematic analysis was conducted using the theoretical framework of social constructivism.”

5.       Ln 21-22: Say that the theme arose during analysis, no need to say secondary, this is a result not method.

6.       Line 23: Do you have any numbers? what portion of students/groups say this?

7.       Line 30-31: This is not an appropriate conclusion/summary statement for your results. Your results do not support this.

Introduction:

8.       Overall the introduction brings in a number of topics that are extraneous to developing the story to justify the importance of the study. It needs to be significantly trimmed down and restructured to better build a coherent case for this study.

9.       Paragraph 1: Introduce MH in the first paragraph to state that it helps understand these changes.

10.   Reduce verboseness in this first paragraph, and throughout the introduction, much of the material here is unnecessary.

11.   Ln 38-40: this sentence is generally awkward needs to be rephrased. For part of it on ln 39-40, rephrase “…and attitudes in understanding the changes...” to say “and attitudes to understand the changes…”

12.   Ln 47-49: Just say there is growing importance for rather than careful observed trends demonstrate.

13.   Ln 51-53: Condense this sentence. You can just say The importance of humanities in medical education has been known since the 1960s (refs), and today MH classes are widespread…

14.   Ln 58-61: simplify, confusing what these means in a practical sense.

15.   Ln 62-67: this seems to tie in with the more classes in previous paragraph and should be joined.

16.   Ln 65-66: delete this sentence

17.   Ln 68-77: Delete this paragraph and figure. You do not discuss integration anywhere in the paper. If you feel this guided your interview guide development or theme development it can be put in the methods.

18.   Ln 81: Social competencies developed by MH?

19.   Ln 81-97: There is a lot going on in the intro. This section seems to be the driver of why this paper is important and it should be emphasized, remove the fluff and condense the first paragraphs. That info can be found elsewhere in other manuscripts and does not need to be in your introduction.

20.   Ln 123-124: the MH thread appearing is results, not methods.

Methods:

21.   Separate methods into sub headings. Something like: Setting (describe pop and sampling), framework (explain what social constructivism is), qualitative analysis (the exact methods of collecting the data and analysing it.

22.   Ln 118: Delete “The qualitative research –“ also use “were” conducted, not “was” conducted.

23.   Ln 118-119: Include # of medical students. Also, How many medical schools in these cities? only three? If only one per city say, students from three medicals schools in the cities of Gdansk, etc.

24.   Ln 120-121: this conflicts with the following statement about interviews focused on pros and cons of studying medicine. You need to better clarify what was in the interview.

25.   Figure 2: delete this figure it is extraneous

26.   Table 1: just summarize in text, this table doesn’t add anything

27.   Ln 138: rephrase this to simplify and say Ethics committee approval was waived for this study.

28.   Ln 139-143: This ethics info does not need to be included.

29.   Ln 144-146: Funding and data availability statements below at the end of the manuscript in the sections the journal defines in acknowledgement section.

30.   Ln 148-154: Reiterate the methods here in more detail. Just referencing the other paper is not appropriate for this type of study, this paper should stand alone. how many coders were there, were the interviews recoded and transcribed?

31.   Ln 149: specify this is principles of ‘Framework for applied policy research’, which consists of a five-stage process including familiarization, identifying themes, indexing, charting, and interpretation. the ref Braun and Clark is not going to be widely recognized

32.   Did you have an established set of hypotheses you investigated or did the themes arise through something like grounded theory? E.g. "As no formal a priori hypotheses had been constructed, development of codes and themes arose from the data, using the principles of grounded theory."

Results:

33.   Ln 164: What is most? Quantify these statements

34.   Ln 169: Stylistic choice, don’t state “here is an example” you can integrate examples into the text more seamlessly.

35.   Ln 173: delete “authors” the brackets already indicate inserted text.

36.   Unless you found patterns in the perceptions based on the participant qualities, remove the gender, med year, city, and group number from the quotes. You do not have any discussion pertaining to these and they are distracting.

37.   Ln 178: confusing sentence

38.   Ln 181-182: delete “(according to them)”

39.   Ln 186: you mention both humanities and MH somewhat interchangeably. Be consistent on terms as they are not exactly synonymous. I recommend choosing one.

40.   Ln 195: Why is this last line included? Doesn’t add anything

41.   Ln 196: change “of” to “to” here and throughout

42.   Ln 197: say theme arose, delete “spontaneously appeared” makes it seem like it wasn't a theme you determined in analysis

43.   Ln 198: is this the disadvantages of studying medicine? Or MH?

44.   Ln 200-201: “in the last group, other questions were asked” What does this mean?

45.   Ln 201-202: The results presented prior to this do not support this statement.

46.   Ln 206: do you mean other courses, not just humanities?

47.   Ln 264: This label doesn’t make sense, what does that mean? Relabel and consider describing the meanings of these themes in the methods

Discussion:

48.   The discussion needs to be restructured as it feels fractured jumping from one topic to another and back, and as a result confusing and does not flow.

49.   The takeaway of this paper needs to be more clearly defined and discussed in the discussion. The paper makes statements that are not supported by the findings and goes on tangents about topics that do not tie into the paper purpose. The discussion points need to be streamlined or developed further to clarify the discussion takeaways.

50.   Unusual is the wrong word. Do you mean it has not been previously performed? or that there is limited literature in this topic?

51.   Ln 268: You emphasize this in multiple places in the discussion, but the fact the moderator did not prompt the discussion is not unique. Unexpected themes often develop from semistructured interviews and focus groups. Minimize these references.

52.   Ln 269: delete “at the beginning of this study”

53.   Ln 295-297: where/how did you identify this? I didn't see this theme in the results. You could say "This divergence in opinion emphasizes the importance of considering the needs and values of medical students when planning MH curricula."

54.   Ln 298: The name Douglas is not consistent with what is in the references

55.   Ln 301: This makes it seem like the qual analysis was separate from the survey. They collected data via a semi-structured survey, you did focus group interviews that should be explained if included.

56.   Ln 302-305: this doesn't seem particularly relevant and could be subsumed in the fact they used a survey and you and interview

57.   Ln 308-310: This is the first mention of this and is stated as a fact. Where is the reference? You do not demonstrate this is the results.

58.   Ln 311-313: delete these two first sentence or at least combine them into one. They are repetitive and do not add relevant substance to the paper. It is known that things like training impacts teaching quality.... its more important to discuss what’s done in Poland.

59.   Ln 318-319: This sentence doesn’t make sense, it is disjointed and seems like a fact, but there is no reference. In Ln 321: you mention that the content taught is essential, to tie these ideas together explain that the students want higher quality teaching of humanities.

60.   Ln 322: put this towards the top of the paragraph to introduce the topic. The paragraph doesn’t flow.

61.   Ln 325-328: Is there a ref for this?

62.   Ln 331 to 337: delete the last sentence and just reference shapiro.

63.   Ln 338: again “invalid offer” this doesn’t make senses

64.   Ln 342: rephrase to “…from as much as 216 hours of training to as little as 15 hours...”

65.   Ln 351-356: this ties into what you discuss above, but feels orphaned here, these sentences could be tied into discussion above

66.   Ln 359-361: You don't mention these two items previously so they seem out of place. There needs to be a more concise discussion of the purpose of humanities courses

67.   Ln 362: Since when is multitasking humanities. Delete

68.   Ln 366-369: You do not need to keep mentioning that this wasn't an originally investigated theme. You can just say more work is needed. Move that to the discussion because “more research needed” isn’t a limitation, but important to discuss.

69.   Ln 368-369: This is not a limitation of qualitative research, the lack of generalizability is not the qualitative fault, but the population you use etc. Rewrite.

70.   Ln 373-378: Why does the knowing each other matter? That isn’t a typical requirement. The not speaking their mind is a limitation of focus groups rather than possibly knowing each other, social desirability bias, personality etc.

71.   Ln 380-381: since when have focus groups not been conducted in Poland? There is no evidence to support this claim. You can state (if true) that this is the first study to use focus groups to examine this topic.

72.   Ln 384: “strongly suggesting” this should be in the discussion but it is not a strength of the study specifically.

Conclusion:

73.   Ln 387: Delete “in conclusion”

74.   Ln 387-390: You did not demonstrate that students spontaneously discuss humanities. The conclusion should discuss your theme results. In general stop emphasizing the spontaneous discussion of humanities. It is irrelevant and uninteresting.

75.   Ln 390-392: This is not the take away of your paper. Remove this. Your paper is about student perceptions of humanities. The take away should be relevant to that such as above in the discussion when you mention the importance of considering student needs and values in developing programs, or that students think humanities are important, but how they are taught discourages students.

Author Response

Dear Reviewer #1,
The authors thank the Reviewer for the time and thoughts spent on this manuscript. The patience and guiding hand of the Reviewer are very much appreciated. The Reviewer’s suggestions and constructive criticism were used to improve this work.

Comment #1
When you say “focused interviews,” are you alluding to the focus group or somethings else? it seems like an un-useful descriptor, regardless, and I would suggest removing.
We followed this suggestion and removed “focused interviews” from the title, which now reads: “Perception of Medical Humanities among Polish Medical Students: Qualitative Analysis”

Comment #2
Clarify that the discussion was benefits of medicine (or pharmaceuticals? –mentioned later is paper) and say the theme of MH and MH classes arose during analysis.
The suggested changes were introduced and the revised passage reads as follows:
“The students attended one of three Polish medical schools in Gdansk, Krakow, and Warsaw and were recruited to the discussion focused on the impact of drug manufacturers’ presence at medical universities on socialization in the medical profession. The theme of MH and MH classes arose during thematic analysis. “

Comment #3
Ln 18: Awkward sentence, phrase sounds like you analysed the students rephrase to “Interviews from seven focus groups were analysed…” or “The perceptions of medical students form seven focus groups were analysed…
The suggestion is appreciated, we revised it as follows:
“The perceptions of medical students form seven focus groups were analyzed. The students were attending one of three Polish medical schools in Gdansk, Krakow, and Warsaw.”

Comment #4
Ln 20-21: Simplify sentence to state “Thematic analysis was conducted using the theoretical framework of social constructivism.”
The sentence was simplified and reads precisely as it as suggested by Reviewer.

Comment #5
Ln 21-22: Say that the theme arose during analysis, no need to say secondary, this is a result not method.
The sentence was revised and it reads now:
“The theme of MH and MH classes arose during thematic analysis”

Comment #6
Line 23: Do you have any numbers? what portion of students/groups say this?
We appreciate this comment and changed the statement as follows:
“In six groups, students thought that MH courses would be helpful in their future medical practice. However, in four groups, different opinion was expressed that MH courses were unnecessary or even "a waste of time".

Comment #7
Line 30-31: This is not an appropriate conclusion/summary statement for your results. Your results do not support this.
We removed that statement.

Comment #8
Overall the introduction brings in a number of topics that are extraneous to developing the story to justify the importance of the study. It needs to be significantly trimmed down and restructured to better build a coherent case for this study.
The Introduction was revised following the comments and suggestion of the Reviewer listed below.

Comment #9
Paragraph 1: Introduce MH in the first paragraph to state that it helps understand these changes.
To address that comment, we added the following opening paragraph in which we define medical humanities:
“The literature defined medical humanities (MH) in various ways [1-3], but all definitions acknowledge their multi- and interdisciplinary character. MH integrate psychology, social sciences, arts, history, ethics, and philosophy into one discipline with the goal of better understanding the human side of medicine [4]. The Association for Medical Humanities (UK) clearly defined the MH’s aims as developing communi-cation skills, ethical concerns, critical and reflective thinking, concern for different as-pects of the human condition, and own professional and personal values resulting in “intersubjective” knowledge of the patient and physician.”

Comment #10
Reduce verboseness in this first paragraph, and throughout the introduction, much of the material here is unnecessary.
We followed this suggestion and reduced and rewritten that paragraph (now the second in the Introduction).

Comment #11
Ln 38-40: this sentence is generally awkward needs to be rephrased. For part of it on ln 39-40, rephrase “…and attitudes in understanding the changes...” to say “and attitudes to understand the changes…”
These lines were deleted and paragraph rewritten.

Comment #12
Ln 47-49: Just say there is growing importance for rather than careful observed trends demonstrate.
See above.

Comment #13
Ln 51-53: Condense this sentence. You can just say The importance of humanities in medical education has been known since the 1960s (refs), and today MH classes are widespread…
We folloled this suggestion and revised as follows:
“The importance of humanities in medical education has been known since the 1960s [5,6] [6][7][8][9], and today, MH classes are widespread, providing theoretical and practical frameworks for integrating the humanities into medicine [10].”

Comment #14
Ln 58-61: simplify, confusing what these means in a practical sense.
We shortened and simplified that sentence. It reads now:
“Contemporary research points to the variety of approaches used to teach MH [12] and supports the positive outcomes of MH teaching during undergraduate studies [13], despite still existing dichotomy between medical skills courses and humanity-oriented classes [14].”

Comment #15
Ln 62-67: this seems to tie in with the more classes in previous paragraph and should be joined.
We merged the paragraphs to keep the flow.

Comment #16
Ln 65-66: delete this sentence
The sentence was deleted.

Comment #17
Ln 68-77: Delete this paragraph and figure. You do not discuss integration anywhere in the paper. If you feel this guided your interview guide development or theme development it can be put in the methods.
The paragraph and the figure were deleted.

Comment #18
Ln 81: Social competencies developed by MH?
Thank you for this suggestion, the sentence reads now:
“Social competencies developed by MH strengthen understanding and reaction to human behavior during the decision-making pathway (e.g., facilitating the diagnosis process)..”

Comment #19
Ln 81-97: There is a lot going on in the intro. This section seems to be the driver of why this paper is important and it should be emphasized, remove the fluff and condense the first paragraphs. That info can be found elsewhere in other manuscripts and does not need to be in your introduction.
The paragraph was shortened, and it reads now:
“Social competencies developed by MH strengthen understanding and reaction to the human behavior of students and medical practitioners during the decision-making pathway (e.g., facilitating the diagnosis process). They also strengthen cooperation skills, positively changing the environment and enhancing the therapy outcomes [20]. MH-enhanced education should prevent professional burnout and moral erosion [21]. As a result, the quality and efficiency of physicians' work improve in many areas of functioning.”

Comment #20
Ln 123-124: the MH thread appearing is results, not methods.
We shifted that to the Results section.

Comment #21
Separate methods into sub headings. Something like: Setting (describe pop and sampling), framework (explain what social constructivism is), qualitative analysis (the exact methods of collecting the data and analysing it.

The comment is appreciated. The MM was split into sections with sub-headings. The framework was expanded, and the definition of social constructivism and examples were added. That passage reads now:
2. Materials and Methods
Setting:
The focus group interviews were conducted in November 2019 and involved medical students from three large Polish cities – Gdansk (which has one medical school), Krakow (two medical schools), and Warsaw (four medical schools). Students from only one medical school per city were recruited. The main objective of that study was to identify the influence of pharmaceutical companies on socialization in the medical profession [see published papers [26,27]); however, a substantial part of the conversation was devoted to socialization in the medical profession in general. To cre-ate a friendly and relaxed atmosphere, the moderator asked a few simple introductory questions, for example, about the pros and cons of studying medicine.
The sampling for the study was purposive. The recruiting company invited sec-ond-year or higher medical students to participate. There were 68 students in the sev-en analyzed focus groups. In Gdańsk, there were 26 persons recruited; in Krakow, 16 (8 women and 8 men); and in Warsaw (four medical schools), 26 (11 women and 15 men).
Framework:
The theoretical framework in this work was social constructivism. It is based on certain assumptions: 1) reality is created by human actions, and it cannot be discovered because it is socially created; 2) knowledge is a human product, and it is socially and culturally created; 3) learning is a social process, and meaningful learning takes place when a person is engaged in social activities [28]. Thus, social constructivism assumes that people construct their knowledge while interacting with others [29]. Things that seem neutral or ordinary in a society, such as a gender and race, are socially con-structed and do not reflect reality. How we perceive the world depends on the lan-guage, culture, or historical epoch [30]. Therefore, the medical students were charac-terized by their unique constructs,
recognized by the research method. Focus groups share some assumptions with social constructivism (the subjects construct their own framework during the discussion to give meaning to their experiences), and in interac-tion with others, this can change, creating new knowledge [31].
Qualitative analysis:
All focus groups lasted approximately 2 hours. They were all run by the same moderator and used the same script. Based on the recordings (video and audio), an anonymized transcript of all focus groups was prepared. Thematic analysis was used to examine the result [32-34]. The thematic analysis involves searching the data set to identify, analyze and report recurring patterns [35] and can be applied to various para-digms. However, it seems particularly well suited to study social constructivism, on which this survey initially focused [36].
As an a priori hypothesis was not developed, codes and themes were developed inductively from the data. The data was coded by the project's PI, who participated in the entire study process and interpreted the context. The thematic analysis followed the steps described in the literature: familiarization, identifying themes, indexing, charting, and interpretation [37,38]
Ethics committee approval was waived for this study.

Comment #22
Ln 118: Delete “The qualitative research –“ also use “were” conducted, not “was” conducted.
The suggested corrections were made.

Comment #23
118-119: Include # of medical students. Also, How many medical schools in these cities? only three? If only one per city say, students from three medicals schools in the cities of Gdansk, etc.
We revised that sentence. It reads now:
“The sampling for the study was purposive. The recruiting company invited second-year or higher medical students to participate. There were 68 students in the seven analyzed focus groups. In Gdańsk (one medical school), there were 26 persons recruited (17 women and 9 men); in Krakow (two medical schools), 16 (8 women and 8 men); and in Warsaw (four medical schools), 26 (11 women and 15 men).

Comment #24
Ln 120-121: this conflicts with the following statement about interviews focused on pros and cons of studying medicine. You need to better clarify what was in the interview.
The interview was described in detail.

Comment #25
Figure 2: delete this figure it is extraneous
As suggested, Figure 2 was deleted.

Comment # 26
Table 1: just summarize in text, this table doesn’t add anything
Table 1 was deleted

Comment # 27
Ln 138: rephrase this to simplify and say Ethics committee approval was waived for this study.
That sentence was rephrased as suggested by the Reviewer.

Comment # 28
Ln 139-143: This ethics info does not need to be included.
That passage was deleted.

Comment # 29
Ln 144-146: Funding and data availability statements below at the end of the manuscript in the sections the journal defines in acknowledgement section.
That passage was deleted.

Comment # 30
Ln 148-154: Reiterate the methods here in more detail. Just referencing the other paper is not appropriate for this type of study, this paper should stand alone. how many coders were there, were the interviews recoded and transcribed?
The methods were expanded and detailed description of how the study/interwievs were conducted added. The interviews were recorded and transcribed (stated in lines 113 – 114). There was one coder, this is now stated.
“All focus groups lasted approximately 2 hours. They were all run by the same moderator, who also acted as coder, and used the same script. Based on the recordings (audio and video), an anonymized transcript of all focus groups was prepared.”

Comment # 31
Ln 149: specify this is principles of ‘Framework for applied policy research’, which consists of a five-stage process including familiarization, identifying themes, indexing, charting, and interpretation. the ref Braun and Clark is not going to be widely recognize
We now provide the following explanation and refer to the book Analyzing Qualitative Data (Alan Bryman, Bob Burgess):
“The thematic analysis followed the steps described in the literature: familiarization, identifying themes, indexing, charting, and interpretation [37,38]. “

Comment # 32
Did you have an established set of hypotheses you investigated or did the themes arise through something like grounded theory? E.g. "As no formal a priori hypotheses had been constructed, development of codes and themes arose from the data, using the principles of grounded theory."
To address this comment, we added the following:
“As an a priori hypothesis was not developed, codes and themes were developed inductively from the data.”

Comment # 33
Ln 164: What is most? Quantify these statements
We revised this statament and write now:
“In all analyzed groups, some students considered medical studies to be complex and requiring much dedication.”
Focus group research deals with the analysis of the overall impression of the discussion. What we can say based on the analysis is that there were no objections to the statements of persons who said that these studies were demanding and difficult.

Comment # 34
Ln 169: Stylistic choice, don’t state “here is an example” you can integrate examples into the text more seamlessly.
We removed that statement and added a colon to the preceeding sentence to illustrate our statement.

Comment # 35
Ln 173: delete “authors” the brackets already indicate inserted text.
“authors” were deleted from all statements.

Comment # 36
Unless you found patterns in the perceptions based on the participant qualities, remove the gender, med year, city, and group number from the quotes. You do not have any discussion pertaining to these and they are distracting.
All of this was removed. Only in one case, conversation of two persons, we added “Person A” and Person B” for clarity.

Comment # 37
Ln 178: confusing sentence
That sentence was removed.

Comment # 38
Ln 181-182: delete “(according to them)”
Deleted.

Comment # 39
Ln 186: you mention both humanities and MH somewhat interchangeably. Be consistent on terms as they are not exactly synonymous. I recommend choosing one.
The comment is very appreciated. We opted for using “medical humainities” or MH throughout the entire manuscript.

Comment # 40
Ln 195: Why is this last line included? Doesn’t add anything
That line was deleted.

Comment # 41
Ln 196: change “of” to “to” here and throughout
Done.

Comment # 42
Ln 197: say theme arose, delete “spontaneously appeared” makes it seem like it wasn't a theme you determined in analysis
That expression was revised as suggested.

Comment # 43
Ln 198: is this the disadvantages of studying medicine? Or MH?
The discussion concerned the disadvantages of studying medicine.

Comment # 44
Ln 200-201: “in the last group, other questions were asked” What does this mean?
That passage was revised and reads now:
“In five of the seven analyzed focus groups, the MH theme arose when the partici-pants discussed the disadvantages of studying medicine (Gdansk, GP 2; Krakow GP 6; Warsaw GP 7, Warsaw GP 8, Warsaw GP 9). In one of the five groups, the theme of disadvantages arose when the advantages of studying medicine were discussed (Kra-kow, GP 5); in another group when other issues were debated (Gdansk, GP 3). The students often associated the MH with something negative. Four subthemes of dis-couragement to learning MH theme were recognized, as described below”

Comment # 45
Ln 201-202: The results presented prior to this do not support this statement
That means that medical students often associate the humanities with something negative.
The sentence was revised. We do not conclude but state the fact:
“The students often associated the MH with something negative.”

Comment # 46
Ln 206: do you mean other courses, not just humanities?
Yes, this was precisely the meaning of that sentence. We removed it because it does not contribute to the meaning of this work.

Comment # 47
Ln 264: This label doesn’t make sense, what does that mean? Relabel and consider describing the meanings of these themes in the methods.
We change the label to “Elective versus compulsory format“. This change apprears now in the M&M and Figure 1.

Comment # 48
The discussion needs to be restructured as it feels fractured jumping from one topic to another and back, and as a result confusing and does not flow.
The discussion was revised according to Reviewers sugesstions.

Comment # 49
The takeaway of this paper needs to be more clearly defined and discussed in the discussion. The paper makes statements that are not supported by the findings and goes on tangents about topics that do not tie into the paper purpose. The discussion points need to be streamlined or developed further to clarify the discussion takeaways.
The discussion was revised according to Reviewers sugesstions.

Comment # 50
Unusual is the wrong word. Do you mean it has not been previously performed? or that there is limited literature in this topic?
We removed the first paragraph from the Discussion.

Comment # 51
Ln 268: You emphasize this in multiple places in the discussion, but the fact the moderator did not prompt the discussion is not unique. Unexpected themes often develop from semistructured interviews and focus groups. Minimize these references.
We removed the first paragraph from the Discussion.

Comment # 52
Ln 269: delete “at the beginning of this study”
We removed that phrase.

Comment # 53
Ln 295-297: where/how did you identify this? I didn't see this theme in the results. You could say "This divergence in opinion emphasizes the importance of considering the needs and values of medical students when planning MH curricula."
The sentence was changed as suggested.

Comment # 54
Ln 298: The name Douglas is not consistent with what is in the references
We replaced the name “Douglas” (which is the middle name of the author, which we mistakenly took for the last name) with “Helen”.

Comment # 55
Ln 301: This makes it seem like the qual analysis was separate from the survey. They collected data via a semi-structured survey, you did focus group interviews that should be explained if included.
We revised that paragraph, taking into account the changes suggested by the Reivewer: “Our first finding of diverse opinions about MH contrasts with the study of Helen et al., in which 98.5% of participating medical students had positive or highly positive opinions about MH [39]. However, there are substantial methodological differences between the study of Helen et al. and ours, as they performed a semi-structured survey, whereas we used a focus group interview.”

Comment # 56
Ln 302-305: this doesn't seem particularly relevant and could be subsumed in the fact they used a survey and you and interview
These lines were deleted, see above.

Comment # 57
Ln 308-310: This is the first mention of this and is stated as a fact. Where is the reference? You do not demonstrate this is the results.
We deleted the two sentences. They were an attempt to interpret some of the results.

Comment # 58
Ln 311-313: delete these two first sentence or at least combine them into one. They are repetitive and do not add relevant substance to the paper. It is known that things like training impacts teaching quality.... its more important to discuss what’s done in Poland.
We merged the two sentences as suggested.

Comment # 59
Ln 318-319: This sentence doesn’t make sense, it is disjointed and seems like a fact, but there is no reference. In Ln 321: you mention that the content taught is essential, to tie these ideas together explain that the students want higher quality teaching of humanities.
The sentence in lines 318 – 318 was deleted and that in line 321 was modified. That passage reads now:
“The quality of teaching depends on the instructor's knowledge and skills and should be controlled by each university using evaluation tools (e.g., post-classes quality assessment) [40,41]. In Poland, training the MH instructors is left to the discretion of individual medical schools. However, in light of our findings, universal introductory MH courses for the academic staff could help shape the teachers. Such training involves considerable costs, which might pose a problem, especially since the state-
founded universities in Poland are still underfunded. Still, the students want higher quality teaching of MH.”

Comment # 60
Ln 322: put this towards the top of the paragraph to introduce the topic. The paragraph doesn’t flow.
We shifted that sentence to the start of the paragraph.
The revised parapgraph reads now:
“Knowledge, competencies, and skills in the humanities and social sciences must consider the person as a human being with his or her problems, capable of self-determination, which is particularly important in the case of professions of public trust, of which the medical profession is one. The contemporary physician cannot only possess technical and practical skills, but it is advisable to develop personal and interpersonal skills to create a relationship with the patient and the patient's family.[40]. The quality of teaching such skills depends on the instructor's knowledge and competencies and should be monitored by each university using evaluation tools (e.g., post-classes quality assessment) [41,42]. In Poland, training the MH instructors is left to the discretion of individual medical schools. However, in light of our findings, universal introductory MH courses for the academic staff could help shape the teachers. Such training involves considerable costs, which might pose a problem, especially since the state-founded universities in Poland are still underfunded. Still, the students want higher quality teaching of MH.”

Comment # 61
Ln 325-328: Is there a ref for this?
We support now our statement with the following reference:
Simpson M, Buckman R, Stewart M, Maguire P, Lipkin M, Novack D, Till J. Doctor-patient communication: the Toronto consensus statement. BMJ. 1991 Nov 30;303(6814):1385-7. doi: 10.1136/bmj.303.6814.1385. PMID: 1760608; PMCID: PMC1671610.

Comment # 62
Ln 331 to 337: delete the last sentence and just reference shapiro.
This was done according to the instruction.

Comment # 63
Ln 338: again “invalid offer” this doesn’t make senses
This has been changed in the entire manuscript, including the Figure 1. We are now writing about the class format issue (elective versus compulsory).

Comment # 64
Ln 342: rephrase to “…from as much as 216 hours of training to as little as 15 hours...”
The passage was rephrased.

Comment # 65
Ln 351-356: this ties into what you discuss above, but feels orphaned here, these sentences could be tied into discussion above
That change has been done.

Comment # 66
Ln 359-361: You don't mention these two items previously so they seem out of place. There needs to be a more concise discussion of the purpose of humanities courses
We have rewritten the discussion and focused on the findings of this work.

Comment # 67
Ln 362: Since when is multitasking humanities. Delete
Deleted.

Comment # 68
Ln 366-369: You do not need to keep mentioning that this wasn't an originally investigated theme. You can just say more work is needed. Move that to the discussion because “more research needed” isn’t a limitation, but important to discuss.
We deleted the passage in question. We also expanded discussion on what is still needed.

Comment # 69
Ln 368-369: This is not a limitation of qualitative research, the lack of generalizability is not the qualitative fault, but the population you use etc. Rewrite.
The sentence was rewritten and reads now:
“A limitation of this study is purposive sampling, which makes generalization impossible. “
Comment # 70
Ln 373-378: Why does the knowing each other matter? That isn’t a typical requirement. The not speaking their mind is a limitation of focus groups rather than possibly knowing each other, social desirability bias, personality etc.
We have deleted this statement and revised the Conclusions.

Comment # 71
Ln 380-381: since when have focus groups not been conducted in Poland? There is no evidence to support this claim. You can state (if true) that this is the first study to use focus groups to examine this topic.
The sentences was revised and reads now:
“The study's strength is the focus group technique, used for the first time in Poland to examine the topic.”

Comment # 72
Ln 384: “strongly suggesting” this should be in the discussion but it is not a strength of the study specifically.
That passage was removed.

Comment # 73
Ln 387: Delete “in conclusion”
The sentence was revised.

Comment # 74
Ln 387-390: You did not demonstrate that students spontaneously discuss humanities. The conclusion should discuss your theme results. In general stop emphasizing the spontaneous discussion of humanities. It is irrelevant and uninteresting.
We stopped emphasizing the issue and rewritten the conclusions.

Comment # 75
Ln 390-392: This is not the take away of your paper. Remove this. Your paper is about student perceptions of humanities. The take away should be relevant to that such as above in the discussion when you mention the importance of considering student needs and values in developing programs, or that students think humanities are important, but how they are taught discourages students.

The comment is appreciated. We have revised the Conclusions as follows:
“5. Conclusion
The students' views on whether medicine and medical studies are related to MH and the need to study MH were different among the respondents. Our study, therefore, indicates the need to demonstrate the importance of MH, e.g., presenting a practical application in future medical practice. The present study determined factors discouraging the respondents from learning humanities – these factors included poor quality of teaching, time investment, allocation to inappropriate years of studies, and unsuitable content or format of the classes. The results of this study may create the basis for further quantitative research on how problematic these issues are in Polish medical universities. Addressing these problems can help improve the training of future physicians in Poland.”

Reviewer 2 Report

Dear authors,

I was impressed with the subject of the article.

I have a couple of observations:

1. the biggest issue seems to be the small sample of students, only 68, which could pose a problem from a statistical perspective, which means that the results are harder to generalize to a greater scale. A bigger sample would be advisable.

2. Although the chapters and sub-chapters are neatly separated and easy to follow, I would advise to add a „Future research”, as a separate chapter

3. Also, consider expanding on the Conclusion section.

Author Response

Response to reviewer 2

The authors thank the Reviewer for the time spent reviewing our manuscript and the creative criticism. Below you will find a point-to-point answer to all queries.

Comment #1

1. The biggest issue seems to be the small sample of students, only 68, which could pose a problem from a statistical perspective, which means that the results are harder to generalize to a greater scale. A bigger sample would be advisable.

The problem related to a small sample (typical for qualitative research) was discussed in the limitation section. In the body text, we also suggested further larger, quantitative research on this issue to better diagnose the issues we raised. Our study creates a platform point for building a quantitative study questionnaire.

2. Although the chapters and sub-chapters are neatly separated and easy to follow, I would advise to add a „Future research”, as a separate chapter

We followed the suggestion of the Reviewer and added the following paragraph in the Discussion:

“Future research Our study created a starting point for conducting quantitative research on this issue at Polish medical universities. First of all, it shows what issues should be raised, thus, facilitating the creation of a survey questionnaire. Furthermore, our study indicated that students from more advanced years should participate in such a survey, as they have more experience with humanities at the university. Two issues highlighted by students (e.g., allocation of MH to inappropriate study years or inappropriate content of format of the classes) suggest a topic for another study, in which analysis curriculum of Polish medical schools could be conducted to determine which type of MH classes and when are taught in Poland to whether the students' criticism was justified. Students also raised the issue of being taught MH by physicians who often are unfamiliar with the subject. On the other hand, they had reservations that people teaching humanities are detached from medical practice. That suggests a focus of another future study on the professional background of MH teachers in Poland.”

3. Also, consider expanding on the Conclusion section.

The section Conclusions was rewritten and reads as follows:
“The students' views on whether medicine and medical studies are related to MH and the need to study MH were different among the respondents. Our study, therefore, indicates the need to demonstrate the importance of MH, e.g., presenting a practical application in future medical practice. The present study determined factors discouraging the respondents from learning humanities – these factors included poor quality of teaching, time investment, allocation to inappropriate years of studies, and unsuitable content or format of the classes. The results of this study may create the basis for further quantitative research on how problematic these issues are in Polish medical universities. Addressing these problems can help improve the training of future physicians in Poland.”

Round 2

Reviewer 1 Report

My major concerns have been address there are a few minor issues that need to be addressed:

Re: previous comment 3: My apologies, I made a typo in the phrasing of my revision. “form” should be “from” in this sentence.

Comment #3
Ln 18: Awkward sentence, phrase sounds like you analysed the students rephrase to “Interviews from seven focus groups were analysed…” or “The perceptions of medical students form seven focus groups were analysed…
The suggestion is appreciated, we revised it as follows:
“The perceptions of medical students form seven focus groups were analyzed. The students were attending one of three Polish medical schools in Gdansk, Krakow, and Warsaw.”

Ln 54 . Make this an “and” not “or”

Ln 87-90: Unless, I’m misunderstanding, you say you only recruited from one school each city? In that case I’d leave out the total number of medical schools in each city. Just say “The focus group interviews were conducted in November 2019 and involved medical students from three large Polish cities – Gdansk, Krakow, and Warsaw.” Same on line 100.

Ln 227: change to “In another group a student stated”

Ln 306-307: Remove “impossible” and replace with “which limits generalizability.”

Author Response

Reply to Reviewer 1

My major concerns have been address there are a few minor issues that need to be addressed:

The guidance of Reviewer 1 is highly appreciated.

Re: previous comment 3: My apologies, I made a typo in the phrasing of my revision. “form” should be “from” in this sentence.

Comment #3
Ln 18: Awkward sentence, phrase sounds like you analysed the students rephrase to “Interviews from seven focus groups were analysed…” or “The perceptions of medical students form seven focus groups were analysed…
The suggestion is appreciated, we revised it as follows:
“The perceptions of medical students form seven focus groups were analyzed. The students were attending one of three Polish medical schools in Gdansk, Krakow, and Warsaw.”

The sentence in Line 18 reads now :  “Interviews from seven focus groups were analysed

Ln 54 . Make this an “and” not “or”

Changed.

Ln 87-90: Unless, I’m misunderstanding, you say you only recruited from one school each city? In that case I’d leave out the total number of medical schools in each city. Just say “The focus group interviews were conducted in November 2019 and involved medical students from three large Polish cities – Gdansk, Krakow, and Warsaw.” Same on line 100.

The sentence in lines 87 – 90 was revised according to the suggestion. The number of medical schools in Warsaw was deleted from line 100.

Ln 227: change to “In another group a student stated”

Corrected.

Ln 306-307: Remove “impossible” and replace with “which limits generalizability.”

Corrected.

Reviewer 2 Report

Congratulations on the work.

I would still expand the conclusion section.

Author Response

Reply to Reviewer 2

Congratulations on the work.

I would still expand the conclusion section.

We appreciate the advice. Since the Conclusion section needs to be concise, we added only one statement, which conveys many ideas:

“Moreover, we advocate a need to create a national MH organization to collaborate with others (e.g., the Association for Medical Humanities in the UK) and to advise Polish medical schools on creating and implementing their MH curricula. This organization should include representatives of advanced medical students to acknowledge their opinions in the future. ”